# Estimation of Flood Inundation Area Using Soil Moisture Active Passive Fractional Water Data with an LSTM Model

**DOI:** 10.3390/s25082503

**Published:** 2025-04-16

**Authors:** Rekzi D. Febrian, Wanyub Kim, Yangwon Lee, Jinsoo Kim, Minha Choi

**Affiliations:** 1Department of Global Smart City, Sungkyunkwan University, Suwon 440-746, Republic of Korea; rekzidf.39@skku.edu (R.D.F.); wanyub@skku.edu (W.K.); 2Major of Geomatics Engineering, Division of Earth Environmental System Science, Pukyong National University, Busan 48513, Republic of Korea; modconfi@pknu.ac.kr (Y.L.); jinsookim@pknu.ac.kr (J.K.); 3Department of Water Resources, Graduate School of Water Resources, Sungkyunkwan University, Suwon 440-746, Republic of Korea; 4School of Civil, Architectural Engineering & Landscape Architecture, Sungkyunkwan University, Suwon 440-746, Republic of Korea

**Keywords:** SMAP, fractional water, LSTM, flood inundation area

## Abstract

**Highlights:**

**What are the main findings?**
High performance of area under the curve (AUC) and confusion-matrix-based evaluation index from an LSTM model.Complex terrain and dense vegetation reduce flood detection performance.

**What is the implication of the main finding?**
A high accuracy estimation of fractional water (FW) in areas near water bodies.Fluctuations in measured FW and producing a false positive (FP) and false negative (FN) detection.

**Abstract:**

Accurate flood monitoring and forecasting techniques are important and continue to be developed for improved disaster preparedness and mitigation. Flood estimation using satellite observations with deep learning algorithms is effective in detecting flood patterns and environmental relationships that may be overlooked by conventional methods. Soil Moisture Active Passive (SMAP) fractional water (FW) was used as a reference to estimate flood areas in a long short-term memory (LSTM) model using a combination of soil moisture information, rainfall forecasts, and floodplain topography. To perform flood modeling in LSTM, datasets with different spatial resolutions were resampled to 30 m spatial resolution using bicubic interpolation. The model’s efficacy was quantified by validating the LSTM-based flood inundation area with a water mask from Senti-nel-1 SAR images for regions with different topographic characteristics. The average area under the curve (AUC) value of the LSTM model was 0.93, indicating a high accuracy estimation of FW. The confusion matrix-derived metrics were used to validate the flood inundation area and had a high-performance accuracy of ~0.9. SMAP FW showed optimal performance in low-covered vegetation, seasonal water variations and flat regions. The estimates of flood inundation areas show the methodological promise of the proposed framework for improved disaster preparedness and resilience.

## 1. Introduction

Many countries worldwide, particularly those with limited resources, have experienced significant economic loss and infrastructure damage due to flooding [1]. Flooding caused by extreme rainfall is one of the most severe natural hazards and is projected to increase in frequency under climate change scenarios [2]. Extreme rainfall events and inadequate storage capacity are the main driving factors in flooding events in various regions. The third heaviest rainfall on record, during the 2023 East Asian rainy season, resulted in severe flooding and landslides across the Republic of Korea [3]. In Lamongan, Indonesia, heavy rainfall in early March 2021, caused the water discharge in the Solo River watershed to rise, affecting numerous villages on Java Island [4]. In Manaus, Brazil, in the Amazon Basin, a high water level of 29.19 m compared to a normal level of 27.00 m in May 2021, after heavy rainfall, resulted in severe flooding along the river [5]. The impact of Tropical Storm Ana triggered widespread flooding, infrastructure damage, and loss of life throughout Malawi on 26 January 2022 [6]. These regions have different topographical characteristics that affect the distribution of flood areas. The large geographical scale of flood-affected regions can lead to underestimations of both the disaster’s impact area and the severity of damage, which hampers immediate relief response activities [7]. To implement preventative measures against flood damage and allocate relief efforts, effective disaster response requires real-time, high-accuracy spatial data on impact zones [8].

To mitigate the constraints of reliance on in situ flood observations, recent literature emphasizes flood estimating and forecasting using satellite data. A range of mitigation strategies has been employed to minimize flood-related damage, along with increasingly advanced tools for detecting flooded areas. Extraction of flooded areas can be performed using satellite data from synthetic aperture radar (SAR), optical sensors, or a combination of these [9,10,11,12]. While optical sensors such as Landsat provide fine resolution (30 m) and SAR systems like Sentinel-1 offer high spatial resolution (10–30 m), both are constrained by relatively long revisit cycles (16 days and 12 days, respectively) [13,14]. Conversely, the Moderate-Resolution Imaging Spectroradiometer (MODIS) delivers frequent daily observations (1–2 days) at moderately fine spatial scales (250 m to 1 km), but its effectiveness is significantly compromised by cloud obstruction during flood events [15,16]. This persistent trade-off among spatial resolution, temporal frequency, and weather penetration capability presents substantial challenges for comprehensive flood monitoring. In such cases, microwave sensors can provide accurate data. Microwave remote sensing enables reliable flood monitoring by overcoming solar, atmospheric, and cloud interference while maintaining high sensitivity to water [17].

Microwave sensors are divided into passive and active types to analyze the distance or surface characteristics of an object [18]. Active sensors emit microwaves and receive the reflected signals, whereas passive sensors detect natural microwaves emitted by objects or the ground [19]. Passive sensors provide higher temporal resolution than active sensors, which is an advantage for flood area detection [20]. Passive microwave sensors use brightness temperature (TB) denoted by the temperature (T) and emissivity (∊) of an object: Tb = ∊T [17]. Due to the different emission properties and thermal inertia of water and land, the observed microwave radiation in general results in lower TB values for water (Tbwater) and higher for land (Tbland > Tbwater).

A Soil Moisture Active Passive (SMAP) satellite equipped with a passive microwave sensor allows global flood monitoring at a coarse ≃36 km spatial scale and high temporal frequency under all weather conditions [21]. The SMAP satellite’s L-band radiometer measures TB in both horizontal (H) and vertical (V) polarizations [22]. The SMAP TB measurements are plotted on a 9 km spatial resolution [23]. The SMAP fractional water (FW) cover generated with 3-day temporal resolution from SMAP TB data has enhanced L-band (1.4 GHz) microwave sensitivity to TB water surfaces and is less affected by atmospheric and vegetation noise than optical/higher-frequency microwave data. Surface FW retrievals derived from SMAP L-band TB can be used for flood estimation [24]. SMAP data effectively addresses key observational limitations in flood monitoring through its unique capabilities. Unlike optical sensors, which are constrained by cloud cover, or SAR systems, which are limited by their orbital configurations and extended revisit cycles, SMAP provides consistent high-frequency observations through its L-band radiometer [25]. This advanced instrument acquires global FW measurements every 2–3 days, offering reliable, all-weather monitoring capabilities. The system’s uninterrupted data stream during flood events represents a significant advancement for hydrological monitoring, overcoming the temporal resolution constraints inherent in both optical and SAR satellite systems.

As climate change intensifies and extreme rainfall risks grow, underscoring the need for precise flood monitoring and forecasting, particularly where data is limited [26]. Accurate monitoring and forecasting of flooding under heavy precipitation are needed to determine the long-term susceptibility of an area to climate change impacts [27]. Modern flood forecasting systems utilize physics-based hydrological models, which quantify the properties of water bodies and are powered by rainfall predictions from numerical weather models (NWP). [28]. Concurrently, NWP products such as precipitation data of The Global Forecast System (GFS) provide a basis for comparison and further refinement of flood estimation accuracy when integrated with a deep learning model. The complementary strengths of diverse remote sensing techniques make multi-sensor data fusion a promising approach for achieving higher spatial resolution, temporal coverage, and accuracy of flood monitoring [29].

Machine learning techniques have yielded promising outcomes for flood estimation in the past few years. Conventional machine learning algorithms, including Support Vector Machines (SVMs) and Random Forest (RF), have demonstrated competence in processing structured spatial data and offer valuable model interpretability [30]. However, these conventional methods exhibit significant limitations in capturing the critical temporal dependencies inherent in flood dynamics. A Classification and Regression Trees (CARTs) model was implemented using Google Earth Engine (GEE) to generate FW forecasts 24 h ahead of the five sub-catchments of the Pungwe Basin [21]. Based on the results of that study, the SMAP, GFS, and Landsat data were used and showed FW estimates showed strong agreement with Landsat-derived inundation extents (R = 0.87; RMSE = 0.68%; nRMSE = 25.6%) for training and validating the CART model. The application of deep learning (DL) methods such as artificial neural networks (ANNs) [31], convolutional neural networks (CNNs) [32], and recurrent neural networks (RNNs) has been increasing mainly due to the large increases in data availability and computing power. These methods employ varied approaches to extract flood patterns from existing data for predicting flood dynamics. A CNN demonstrated exceptional capability in processing spatial data through its hierarchical feature extraction architecture [33]. These networks employ convolutional operations to systematically identify and learn spatial patterns in satellite imagery. However, while CNNs have demonstrated robust performance in static spatial analysis of flood patterns, their fundamental architecture is constrained by an inability to effectively capture and model temporal dependencies and sequential relationships in hydrological data [34]. Among the techniques, RNN had high performance in periodically modeling sequential data with specialized recurrent units [35]. A long short-term memory (LSTM) RNN has been used for flood forecasting to address the inherent limitations of RNN problems of gradient instability, manifesting as vanishing or exploding gradients during training that make it difficult to analyze the long-term dependency learning [36]. When employing DL methods, gradient attenuation posed a persistent challenge, becoming especially severe during backpropagation through lengthy temporal sequences. The issue caused minimal weight updates, stalling learning progress entirely. LSTMs inherently capture and preserve long-range sequential patterns through their gated cell structure, making them ideal candidates for solving such problems and achieving impressive model performance [37].

In this study, the LSTM algorithm is employed to estimate flood inundation areas at each site with a fine-scale 30 m resolution based on derived FW data from SMAP satellite observations. Rainfall forecast data can be used as an input and main predictor in the modeling process with NWP data and a digital elevation model (DEM) product for determining the flood inundation area. Then, synergetic data-driven information is fused using an LSTM model to estimate flood inundation areas with a variety of topographic features. The model was trained with day+0 data and its performance was tested using day+3 flood area-based FW data to evaluate its potential for long-term assessment by adjusting input data on each modeling date. The model was validated with water-masking data from a Sentinel-1A satellite.

## 2. The Study Area

The study area comprises regions prone to recurrent flooding and is characterized by diverse topographical features that influence water flow and flood dynamics, including low-lying floodplains along major rivers. The flood model using the LSTM method based on FW data was tested by processing four flood areas with a variety of topographic features. The Solo River region in Indonesia (Figure 1b) and the Amazon River region in Brazil (Figure 1c) have uniform flat lowland characteristics. Wetlands and agricultural lands in those areas act as temporary water storage areas but can overflow during severe rain events. The Goesan Dam region in the Republic of Korea (Figure 1a) and the Shire River region in Malawi (Figure 1d) have both lowland and highland areas. Adjacent to the floodplains are hilly and mountainous areas where steep slopes contribute to rapid runoff and flash floods in downstream regions.

Large modeling areas are used at the Brazil and Malawi sites, while narrower modeling areas are used at the Indonesia and Republic of Korea sites. The processing areas of the input data and the dates of the flood occurrences are provided in Table 1.

## 3. Materials

### 3.1. SMAP-L4 Surface Soil Moisture

The L-band passive microwave radiometer SMAP satellite has a revisit cycle of three days and provides surface soil moisture (SSM) estimates at ≃40 km resolution. SMAP level 4 (L4) SSM data are obtained using the SMAP TB in the National Aeronautics and Space Administration (NASA) Catchment Land Surface Model (CLSM) with an ensemble Kalman filter. This framework simulates soil water transfer between surface layers and deeper root zones, generating global SSM and root zone moisture fields at enhanced 9 km spatial resolution from https://search.earthdata.nasa.gov/search (accessed on 1 March 2025) [38]. These data were used to characterize baseline soil moisture conditions across the flood-prone study areas [24].

### 3.2. SMAP Fractional Water

SMAP FW can be derived from the enhanced Level-1C (L1C) product with 9 km spatial resolution from https://search.earthdata.nasa.gov/search [39], which contains calibrated and geolocated TB values acquired by the SMAP radiometer during descending and ascending half-orbit passes. FW serves as a critical flood monitoring parameter, quantifying water area dynamics through the following equation:(1)$$FW=\frac{T_{B}^{obs}-T_{B}^{land}}{T_{B}^{water}-T_{B}^{land}}$$
where FW represents the fractional water coverage within the sensor’s footprint instantaneous field of view (IFOV), *TB_water_* represents the TB from water, and *TB_land_* represents the TB from the non-water part of the IFOV [40].

### 3.3. Sentinel-1 Water Mask

Launched for uninterrupted global earth surface monitoring, the Sentinel-1 satellite carries a C-band synthetic aperture radar operating at 5.405 GHz [41]. The satellite provides VH-HV and VV-HH for cross/co-polarization. Sentinel-1 Ground Range Detected (GRD) IW mode Level 1 images (obtained from https://search.earthdata.nasa.gov/search accessed on 1 March 2025) were used to validate the estimated flood area of the LSTM model using a threshold-assisted manual water mask constructed using Sentinel-1 data.

### 3.4. ASTER-GDEM

The Advanced Spaceborne Thermal Emission and Reflection-Global Digital Elevation Map Version3 (ASTER-GDEM v3) dataset from https://search.earthdata.nasa.gov/search (accessed on 1 March 2025) [42] representing topographic features (digital surface model) of the earth was generated from 1.5 million scenes of the L1A product from the ASTER instrument. GDEM v3 covers Earth’s land surface between 83° N and 83° S with a spatial resolution of 1 arc-second (≃30 m) and a coverage of about 99% of Earth’s land areas. The dataset was analyzed to identify key flood-influencing variables and to prepare topographic maps displaying the elevations and slopes of the study areas [43].

### 3.5. GFS Precipitation

The Global Forecast System (GFS) is a numerical weather prediction (NWP) model that generates atmospheric forecasts with a spatial resolution of 0.25° × 0.25° (≃27 km) obtained from the National Centers for Environmental Prediction (NCEP) and generated from Google Earth Engine (GEE) https://developers.google.com/earth-engine/datasets/catalog/NOAA_GFS0P25 (accessed on 1 March 2025). GFS model output variables such as accumulated precipitation, which has a three-hour forecast interval with daily run times of 00.00, 03.00, 06.00, 09.00, 12.00, 15.00, 18.00, and 24.00 UTC. These data were used as predictors for flood inundation forecasting and to describe background rainfall forecast conditions [44].

## 4. Methods

### 4.1. Data Pre-Processing

The most used interpolation algorithms today include nearest neighbor, bilinear, and bicubic interpolation, each offering varying levels of accuracy and image quality. Nearest neighbor interpolation is the simplest method, approximating the sinc function in a two-dimensional image. It determines the pixel value of an insertion point by comparing the distances between four surrounding pixels and selecting the nearest one. While computationally efficient, this method often results in blocky artifacts and lower image quality. In contrast, bilinear interpolation improves accuracy by utilizing four surrounding pixels, and applying linear interpolation both vertically and horizontally. This approach generates more continuous outputs than nearest neighbor interpolation but still has limitations in accuracy, particularly in high-detail applications. For even greater precision, bicubic interpolation employs sixteen surrounding pixels, significantly enhancing both accuracy and image quality [45]. By considering a larger neighborhood of pixels, bicubic interpolation reduces artifacts and produces more natural transitions, making it preferable for high-resolution image processing and spatial data interpolation.

The spatial resolutions of SMAP L4 (≃9 km), SMAP L1C (≃9 km), and GFS (≃27 km) are relatively low. To be processed in the LSTM neural network and obtain flood inundation area from FW data, the datasets need to be adjusted to a set resolution and pixel area [46]. The data was resampled to 30 m resolution to resolve at equivalent spatial scales of ASTER-GDEM using a bicubic interpolation algorithm (Figure 2a) that is rapid, simple to perform, and demonstrates optimal recovery performance of known models. This algorithm is included in several software packages (e.g. PyTorch version: 2.3.1+cpu), is commonly used in engineering scope, and has excellent research importance [47]. Bicubic interpolation uses the weights *w*(*i*,*x*) and *w*(*j*,*y*) calculated for the respective coordinates *x* and *y*. The intensity value at position (*x*,*y*) is calculated as:(2)$$I\left(x,y\right)=\sum_{i=-1}^{2}\sum_{j=-1}^{2}w\left(i-x\right)\cdot w\left(j-y\right)\cdot I\left(x_{0}+i,y_{0}+j\right)$$
where *I*(*x*,*y*) denotes the pixel intensity at coordinates (*x*,*y*) in the upscaled image, *x*_0_ and *y*_0_ are the nearest pixel positions in the original image, *w*(*i*,*x*) and *w*(*j*,*y*) are the calculated cubic interpolation weights, and *I*(*x*_0_*+i*, *y*_0_*+j*) are intensity values at integer grid positions of the 16 nearest neighbors.

### 4.2. LSTM Neural Network

The LSTM architecture includes an input gate, forget gate, and output gate [48], as illustrated in (Figure 2c). These components determine whether to retain or transmit the incoming data, overcome the vanishing gradients problem, and output results from the model. The forget gate (*f_t_*) controls the retention or discarding of prior outputs based on the equation:(3)$$f_{t}=\sigma \;(W_{f}\;x_{t}+U_{f}\;h_{t-1}+b_{f})$$
where the sigmoid function is symbolized as *σ*, the weight matrix from *W_f_* and *U_f_*, the current input vector as *x_t_*, *h_t_*_−1_ stands for the previous layer’s computed output, and *b_f_* is the bias vector.

The input gate (*i_t_*) accepts or discards new information based on(4)$$i_{t}=\sigma \;(W_{i}\;x_{t}+U_{i}\;h_{t-1}+b_{i})$$
where the weight matrices are *W_i_* and *U_i_*, and *b_i_* is the bias vector. The variable *C**_t_* utilizes the tanh function to determine the significance of the pass-through value, as follows:(5)$$\tilde{C_{t}}=tanh\;(W_{c}\;x_{t}+U_{c}\;h_{t-1}+b_{c})$$
where the weight matrices are *W_c_* and *U_c_*, and *b_c_* is the bias vector. Memory (*C_t_*) contains the latest processed output:(6)$$C_{t}=f_{t}\;\cdot \;C_{t-1}+i_{t}\;\cdot \;\tilde{C_{t}}$$
where the memory from the previous unit is *C_t_*_−1_. The concluding phase comprises two distinct components. After generating *h_t_*_−1_ and *x_t_* into the network, the output gate (*O_t_*) releases the new information based on *σ*. Subsequently, the activation output produced by the tanh layer from *C_t_* is multiplied by the output gate.(7)$$o_{t}=\sigma \;(W_{o}\;x_{t}+U_{o}\;h_{t-1}+b_{o})$$(8)$$h_{t}=o_{t}\cdot \;tanh(C_{t})$$
where the weight matrices are represented as *W_o_* and *U_o_*, *b_o_* is the bias vector, and *h_t_* is the output from the present unit of the LSTM model.

### 4.3. Modeling Process of LSTM

Each input dataset of the proposed LSTM model is shown as a single-band image of size width × height; a multi-band image is formed by stacking all layers together (Figure 2a, Table 1). In the multi-band image, each image patch is created by pixel-by-pixel extraction and comprises all pixel vectors, as shown in the feature engineering step (Figure 2b). In the LSTM construction step, the sorted vectors from sequential data are phased and sent to the LSTM workflow to produce the result (Figure 2c). Useful information for the flood inundation estimate can then be passed or stored in hidden layer states of the LSTM model. The LSTM forget gate will discard redundant and irrelevant information. As the spatial variability of flood inundation is based on topography, soil, rainfall, and vegetation, the pattern of flood inundation caused by rainfall depends on soil infiltration and the mechanism of surface runoff [24]. The 2D image data is processed at the pixel level, preserving spatial structure by extracting multi-layer feature values for each pixel and effectively transforming the image into a pixel-wise structured dataset. This approach enables independent pixel analysis while maintaining spatial integrity across raster layers, ensuring robust handling of complex multi-dimensional environmental data. Raster-based geospatial data are restructured into sequential samples, where each pixel’s values are processed alongside its neighbors within a moving window and identifying temporal patterns. Batch normalization is applied to standardize inputs, enhancing model stability and reducing regional biases across varying conditions. This approach allows LSTM to effectively learn spatial interactions, improving generalization and predicting accuracy for flood occurrence at the pixel level.

In the LSTM modeling process, with an aim to minimize loss, the choice of training strategy and hyperparameters critically affects model efficacy. Tanh activation functions were used, a hidden size of 25 was applied, batch sizes of 500 were utilized, the optimizer was set to Adagrad, the loss function was set to MSELoss, and a learning rate of 0.002 was implemented for training from a combination of several sets. An early stop was applied if the development score did not increase for more than 10 training epochs, hyperparameter optimization identified the combination that produced a minimal loss, and the highest development score is considered the best. Appendix A lists the optimized results for the LSTM model, where hyperparameters were manually set. The implemented LSTM architecture comprised an input layer, a 25-cell LSTM hidden layer, and an output layer. The LSTM model first processes FW as input and output target variables and treats geospatial raster data from DEM, SSM, and meteorological variables, representing various environmental and hydrological as predictors. These raster datasets are normalized using min-max scaling to ensure a comparable scale from all features, preventing dominance by large-value variables. Reformats pixel-based data into sequential input samples, ensuring that the LSTM can capture temporal dependencies and learns temporal patterns from the geospatial predictors to predict floodwater presence at each pixel. The number of pixels (n) from Figure 2 that are processed is equal to the number of values width x height in Table 1. To stabilize layer input statistics, the network processed training data in batches and incorporated batch normalization preceding each activation layer. To test the LSTM flood modeling results, the day+3 flood estimates were processed using the trained model to determine the change in flood inundation area (Figure 3a). The model trained with flood condition data from day+0 was applied to the peak flood condition at day+3 (Appendix A).

Flood events often exhibit rapid onset and recession, occurring on timescales shorter than three days. The SMAP dataset may fail to capture critical flood evolution stages, leading to incomplete event representation. Models trained on such data may struggle to accurately predict extreme flood events due to missing key transitional phases. Sequential models, such as LSTM, require consistent time intervals between observations for optimal performance [49]. However, the non-uniform time gaps in SMAP data disrupt the sequential dependencies these models rely on, necessitating data imputation or augmentation techniques to maintain temporal coherence. Without appropriate handling, these discontinuities may degrade model learning and prediction reliability. Furthermore, short-duration flood events, which may last only a few days, are under-represented in SMAP observations due to the 3-day revisit cycle [24].

The rationale for using day+0 data for training and day+3 for testing is based on the 3-day temporal resolution of SMAP FW data. SMAP FW provides updates every three days, meaning each observation represents conditions over a multi-day period rather than an instantaneous snapshot. Training on day+0 ensures that the model learns from the most recent and directly observed flood conditions, while testing on day+3 evaluates the model’s predictive capability for future conditions within the constraints of SMAP’s observational cycle. However, flood dynamics can change rapidly and potentially introduce temporal inconsistencies. These inconsistencies must be carefully considered, as flood extents may evolve significantly between SMAP observation windows, impacting model performance.

### 4.4. Validation

Flood inundation estimates were validated using confusion matrix-derived metrics. FW estimates from the LSTM model were generated using a water/non-water classified image, while the observed value was the inundation area calculated from SAR images. The overall accuracy of the inundation area model (accuracy), the false positive performance (precision), the true positive performance (recall), the comprehensive performance resulting in both false negatives and false positives (IoU), and the balance between precision and recall (F1 score) were used for statistical analysis for day+3 results (Figure 3b). Statistical measures were calculated using the formulas below (Equations (9)–(13)). All indicators fall within a range of 0 to 1, where values approaching 1 signify accuracy in identifying the flood inundation area.(9)$$Accuracy=\frac{TP+TN}{TP+FP+FN+TN}$$(10)$$Precision=\frac{TP}{TP+FP}$$(11)$$Recall=\frac{TP}{TP+FN}$$(12)$$IoU=\frac{TP}{TP+FP+FN}$$(13)$$F1\;\mathrm{score}=2\times \left.\left(\frac{Precision\times Recall}{Precision+Recall}\right.\right)$$
where TP is a true positive; TN is a true negative: FP is a false positive; and FN is a false negative.

Even after a thorough search of the available references, a comprehensive search revealed no established FW-based grading system for surface inundation extent. The classification in this study was determined solely through exploration data assessment. The classification of inundation based on FW values is defined as follows: FW values below 0.15 indicate areas unaffected by flooding, while values of 0.15 or higher denote areas affected by flooding. This classification is derived from preliminary statistical analysis, as there is no widely recognized system for grading inundation levels using FW [50].

## 5. Results and Discussion

### 5.1. Comparison of Modeled and Observed Flood Inundation Areas

Figure 4 shows the distribution of flood areas at four sites. The model results successfully estimated the flood inundation areas and permanent water bodies in Indonesia, Korea, and Brazil, but do not optimally display the water body area in Malawi. The flood coverage area in the model generally was overestimated compared to the observations from the Sentinel-1 water mask. The modeled regions of Indonesia, Korea, and Brazil showed calculated flood coverage values of 100%, 69.17%, and 63.40% in the model compared to observed values of 92.20%, 58.33%, and 56.60%, respectively. In contrast, in Malawi, the flood coverage value was slightly underestimated (42.30%) relative to the Sentinel-1 observation (43.33%).

Areas marked in dark blue indicate areas of flood inundation around streams and in flat areas with low elevation and slope that are prone to flooding [51]. In the Korea, Brazil, and Malawi sites, there are variations in elevation, and the model well displays the possibility of flooded areas around permanent water bodies based on the predictor and elevation calculations. In contrast, the entire Indonesian site is classified as flooded because the model processes all areas of the region as low and flat.

The spatial resolutions of SMAP SSM (≈9 km), SMAP FW (≈9 km), and GFS precipitation (≈27 km) were resampled to match the ASTER-GDEM spatial scale (30 m) using the bicubic interpolation method. This approach facilitated the generation of finer-scale (30 m) FW value predictions in the model, significantly influencing the results. The downscaling process was critical for capturing localized hydrological dynamics, as high-resolution soil moisture, rainfall condition and floodwater estimates are essential for accurate flood forecasting and water resource management in heterogeneous landscapes. The LSTM demonstrated the capability of data-driven approaches for fine-resolution flood inundation modeling using complementary NWPs and satellite-derived data to serve as critical predictors. The estimation of flood inundation areas is significantly influenced by the DEM after targeting the FW value as a reference flood area value. In addition, inundation persistence at a given site is primarily governed by the topographic elevation relative to the adjacent water body, whereas the effects of individual sequential data (SSM and precipitation) during LSTM model processing will be averaged at the domain level where inundation coverage is calculated [46]. This highlights the advantage of unified and spatially varied outputs as it can identify different site interdependencies and sensitivities.

### 5.2. Validation of Flood Inundation Area

To validate flood inundation area from the LSTM model, this section evaluates the agreement between LSTM-derived FW extents and Sentinel-1 SAR water masks through comparative analysis (Figure 5, Table 2). Whereas the Malawi site has the lowest accuracy at 0.74, the Indonesia site has the highest model accuracy at 0.91. This high accuracy suggests that the model performs well overall in distinguishing between positive and negative classes. The highest precision value of 0.92 at the Indonesia site indicates that the positive estimations of the model are correct and demonstrate a strong ability to avoid false positives. In contrast, the Korean site has the lowest precision value at 0.81, indicating that many false positives were detected in the model results compared to the observations. The high recall score suggests that the model responds strongly to positive instances. The recall scores >0.9 in the Korea, Brazil, and Indonesia sites mean that the model correctly identifies the majority of all actual positive instances in each area.

A 0.74 IoU score reflects strong alignment between modeled and actual flood zones in Malawi. The score shows a good degree of accuracy in identifying the location of positive instances, signifying that the model estimations closely match the actual instances. However, the score is not as well-predicted as the results of the Indonesian site model. In contrast to the IoU, the F1 score is particularly useful in evaluating models on datasets with class imbalance. The average F1 score of 0.90 indicates that the model performs exceptionally well in the context of both recall and precision. The model demonstrates a strong balance between identifying positive instances and maintaining high estimation quality.

In general, the LSTM flood inundation modeling based on SMAP FW data shows high-performance estimated results in areas near rivers and in flat areas with low elevation and slope. This is evidenced by the results of the confusion-matrix-based evaluation index, where the regions of Indonesia and Brazil have a validation value of ~0.9 on all indicators (accuracy, precision, recall, IoU, and F1 score). In contrast, Korea and Malawi have values <0.9 on all indicators, indicating that the model is not sufficient for processing flooded areas with a variety of regional characteristics (clear topographic differences). The correlation results show that elevation variance has a strong relationship with the FP rate in all sites, with a correlation coefficient of 0.82. This indicates that more complex terrain, with greater elevation variance, contributes to an increase in false positives and leads to lower accuracy. However, the weaker correlation between elevation variance and FN rate (0.21) suggests that other factors, aside from elevation, may play a more significant role in causing false negatives. Therefore, while terrain complexity has a significant impact on FP, the causes of FN are affected by land cover type and other complex environmental factors in addition to elevation.

Since a larger pixel size results in a larger overestimation of flooded areas [52], enhancing spatial resolution represents a validated approach for improved flood mapping accuracy. This study [53] demonstrated higher-resolution data enables analysis of microscale process-form relationships. Numerous methodologies for integrating multi-factor spatial analyses have been developed in recent decades, including data-driven statistical, knowledge-driven qualitative and quantitative, machine and deep learning, and multi-model ensemble prediction. However, these methods cannot provide information about the relative impact of individual factors on the spatial analysis of the flood inundation area. The presented approach showed elevation, slope, surface soil moisture, and precipitation as dominant factors, exclusion of which significantly reduced the flood inundation areas.

### 5.3. Model Performance and Uncertainty

In this study, as shown in Figure 6, The average area under the curve (AUC) that plots sensitivity (*y*-axis) and 1-specificity (*x*-axis) from the LSTM test set was 0.93, indicating high performance (>0.9) for flood inundation estimation [46]. The ROC curve analysis of the LSTM model across Korea (AUC = 0.90), Indonesia (AUC = 0.96), Brazil (AUC = 0.88), and Malawi (AUC = 0.99) demonstrates strong classification performance, with particularly high accuracy in Malawi and Indonesia. While Korea and Brazil show slightly lower AUC values, they still indicate robust predictive capabilities; this variation is likely attributable to regional topographic heterogeneity, hydrological variability, and dataset quality. Brazil’s diverse flood-prone landscapes and Korea’s varied elevation may introduce additional challenges in feature learning. Despite these variations, the model maintains stable performance across all regions, achieving a favorable balance between true positive and false positive rates. The absence of an abrupt threshold where performance drops significantly suggests reliable generalization across different hydrological conditions. However, for applications requiring precise trade-offs between sensitivity and specificity, further threshold tuning may be necessary. Overall, the consistently high AUC values underscore the model’s reliability for cross-regional flood prediction. The other datasets and algorithms with the LSTM model overestimated the results relative to the validation data. Expansion of flood inundation areas from these results needs to be investigated to determine the causes and methods to overcome its uncertainty.

Almost all sites in this study showed a larger flood area with the LSTM model compared to the Sentinel-1 SAR result (Figure 4). The presence of surface water will cause a high bias in SMAP soil moisture data. Using an inaccurate soil moisture value will introduce uncertainty into the fractional water inundation estimation, and quantifying this uncertainty is difficult. However, if the influence of soil moisture is ignored, fractional inundation in non-flooded areas will be overestimated, causing false alarms on inundation maps [54]. Therefore, the importance of soil moisture as an input in the inundation map is emphasized, even though this leads to an overestimation of FW inundation during floods. The referenced study [24] successfully integrated a dataset for flood monitoring but did not quantify how errors from the original SMAP data propagate through the interpolation process into 30 m, potentially affecting the accuracy of downscaled flood maps. While empirical interpolation improves spatial resolution, it may introduce unaccounted-for uncertainties. The challenges primarily originate from fundamental limitations in current methodologies, particularly the non-linear distortions introduced by conventional interpolation techniques and the critical absence of high-resolution validation datasets. Typically, DEMs serve as the primary data source for generating multiple flood modeling input parameters [55]. A slight failure of the model in detecting the presence of a permanent water body (river flow) occurred at the Malawi site (Figure 4) and is evidenced by the large FN value in the region (Figure 5). This occurs because the detected flood area does not include a clear distinction between the river and the surrounding area, leading to underestimated results. These findings demonstrate that ASTER-GDEM applications in low-lying terrain necessitate systematic correction of elevation-dependent biases in observed data [56]. Thus, the accuracy of DEM-derived inputs directly determines the reliability of flood inundation mapping outputs.

A key concern is the lack of quantification regarding how errors in coarse-resolution SMAP FW data propagate through empirical interpolation into higher-resolution (30 m) flood maps. This omission creates uncertainty in assessing the true accuracy of downscaled outputs, as unaccounted-for distortions may artificially expand flood extents. Additionally, elevation-dependent biases in widely used DEMs, such as ASTER-GDEM, further compromise model performance particularly in low-lying areas, where uncorrected terrain errors lead to underestimation of water bodies and distorted flood delineation. Collectively, these findings underscore the need for enhanced error-tracking frameworks in interpolation processes, DEM bias-correction algorithms, and validation against high-resolution ground truth to reconcile precision with operational flood modeling needs.

For high-dimensional parameter spaces, a reduced set of dominant factors can often be efficiently identified [57]. Since LSTMs are especially well-suited for flood forecasting because these models effectively analyze temporally correlated sequential data, it will be easier to estimate flooding and inform more effective decision-making strategies using these forecasts [58]. Given the nonlinear complexity of flood estimations using inundation models and the critical role of multivariate inputs examined in this study, LSTMs effectively capture nonlinear relationships across multi-source datasets with heterogeneous spatiotemporal resolutions. The LSTM model works well for characterizing the spatiotemporal dynamics of flood events by capturing long-term connection in the data [59]. Further research is needed with additional high-spatial-resolution datasets that relate to the process of flooding and have different values over short periods (<3 days). Additional work could combine this process with other DL models to enhance the performance of LSTM model results.

The 3-day forecast window is crucial in emergency response because it provides enough time for preparedness while maintaining forecast accuracy. If the window is too short, such as less than 24 h, authorities may struggle to issue timely warnings, and large-scale evacuations could become chaotic or even impossible. On the other hand, forecasts extending beyond five days tend to be less reliable, leading to wasted resources and inefficient planning. By striking the right balance, a 3-day forecast allows emergency teams to mobilize resources, issue alerts, and implement safety measures effectively, ensuring communities are well-prepared while minimizing unnecessary disruptions.

### 5.4. Elevation and Land Use/Land Cover (LULC) Impact on Flood Inundation Area

The LULC, including trees, rangeland, crops, and built areas, at lower elevations, impacts the frequency and severity of flooding [60,61]. These results are consistent with the increased flood potential in areas of reduced trees on sloping surfaces, leading to increased runoff to lower surfaces [62]. This can be seen in the northern area of the Korean site (Appendix A and Figure 7), where the flood is concentrated in the low crop and built areas surrounded by the higher tree area. Both model results and observations show flood inundation in the area.

The probability of flooding was higher in lower elevations that were comparatively flat, where there were more crops and built areas. That condition led to an increased vulnerability of communities, assets, and infrastructure networks. These results corroborate existing research that land-use conversion from agriculture and forest to other cover types in downstream regions directly elevated runoff volumes to impermeable built surfaces [63]. As can be seen in (Appendix A), Indonesia and Malawi flood-prone areas were located at lower elevations of comparatively flat cropland and built areas near the river [64]. In contrast, the Brazil site (Appendix A) is wet throughout the year [65]. The southern part of the river is comprised dominantly of trees and flooded vegetation at low elevations and is vulnerable to overflow from the river compared to the higher northern areas.

Figure 7 shows the distribution of FP and FN values for Korea, Brazil, and Malawi sites, incorrectly indicating mostly trees and rangeland. This is because the LSTM model overestimated the flood inundation area as described in Section 3.3. Variability in vegetation height and density can absorb and scatter the L-band microwave signals used by SMAP and affect brightness temperature through emission, scattering, and attenuation, complicating signal interpretation for soil moisture estimates [66]. Surface roughness effects as in the Indonesian site are caused by variations in the reflection of microwave signals due to features such as crops or built area [67], introducing fluctuations in measured FW and producing a large FP result in the area. In addition, heterogeneity in vegetation characteristics (size, morphology, and species composition) can introduce variability in Sentinel-1 SAR-based flood detection accuracy. Flooded vegetation typically exhibits elevated backscatter in SAR imagery owing to strengthened double bounce scattering effects [68]. This effect decreases the detection of shorter wavelengths, such as the C-band used by Sentinel-1 [69], and affects the results of the model.

Figure 8 shows the distribution of FP and FN values for the sites, each at a different elevation. Steep slopes and significant changes in elevation within an observation area can influence measured brightness temperatures [70] used to identify FW areas from SMAP data. Complex terrains, such as mountain ranges and valleys, affect the distribution of microwave signals and complicate data interpretation. The strong influence of elevation variability and the complex correlation between LSTM model data from different sources led to large FP, as in the Korean sites, especially at the water body boundaries at an elevation of 200–300 m. For the Indonesia and Brazil sites, the large FP values at elevations 1–20 m are due to the lack of topographic variability, resulting in overestimated flood inundation areas compared to Sentinel-1 SAR observations. The large FN value at elevations of 1–20 m in the Malawi site indicates inaccuracy of the model estimate of flood inundation area in comparison with validation observations from Sentinel-1 SAR. The underestimated flood extent is due to the challenging flood detection in the area based on the lack of clear distinction between the river and surrounding terrain. To address this, systematic biases can be assessed and mitigated by incorporating alternative satellite-derived DEMs or ground-based observations. Consequently, the quality of the DEM used in modeling plays a crucial role in the accuracy of flood inundation maps [51].

Elevation and LULC were particularly influential on flood dynamics, with lower elevations and impervious landcover surfaces more prone to flooding. These variables also shaped the temporal patterns of inundation, underscoring their critical role in long-term assessments. Results highlighted the sensitivity of flood estimation to environmental characteristics and the importance of selecting relevant input variables. Also, the accuracy of the output may depend on the hydrogeological and topographical variations of the study area. Despite inherent uncertainties, this approach can be applied to other areas. Future studies could optimize input data quality, explore additional variables, and enhance the model’s adaptability to diverse topography and climates.

## 6. Conclusions

Flood inundation modeling using LSTM DL based on SMAP FW data reliably tracks surface water variations under extreme precipitation conditions, indicating the potential utility of this flood estimation technique to inform disaster assessment. FW and soil moisture information obtained from SMAP, in addition to GFS rainfall forecasts and floodplain topography from ASTER-GDEM are combined to create an LSTM DL approach for effective flood inundation area estimation. By resampling the data onto a 30 m grid consistent with DEM data, the LSTM method achieved satisfactory values with high performance (average AUC = 0.93). This is reinforced by the confusion-matrix-based evaluation index (accuracy, precision, recall, IoU, and F1 score) used to validate the flood inundation area results, which had a value of ~0.9 on most indicators. The SMAP FW product exhibits optimal performance in low-covered vegetation regions due to minimal interference, allowing for more accurate surface water fraction detection. Its effectiveness is further enhanced in areas with seasonal water variations, where the contrast between wet and dry conditions improves sensitivity to hydrological changes, particularly in flat terrains that reduce topographic signal distortion. Flood-prone areas were consistently observed in low-lying regions, with the flood extent heavily influenced by local LULC patterns, such as built areas and croplands. Complex terrain features with variability in vegetation density had limitations that led to the overestimation of flood inundation areas. Despite these limitations, the proposed LSTM method is a tool for decision-makers to estimate flood inundation areas. Using the results of this study as guidelines will facilitate water resources management and flood prevention. Exploration of more representative data and/or feature engineering with DL methods can effectively characterize flood dynamics for other advanced models. Since the current LSTM model uses only a single time step, its ability to capture long-term trends is limited. Enhancing the model with multi-step sequences, changing FW data using finer spatial resolution data from other L-band satellites (CYGNSS) with a spatial resolution of 3 km to reduce the uncertainties, incorporating external hydrological data, or integrating convolutional layers (CNN-LSTM hybrid) could further improve predictive performance.

## Figures and Tables

**Figure 1 sensors-25-02503-f001:**
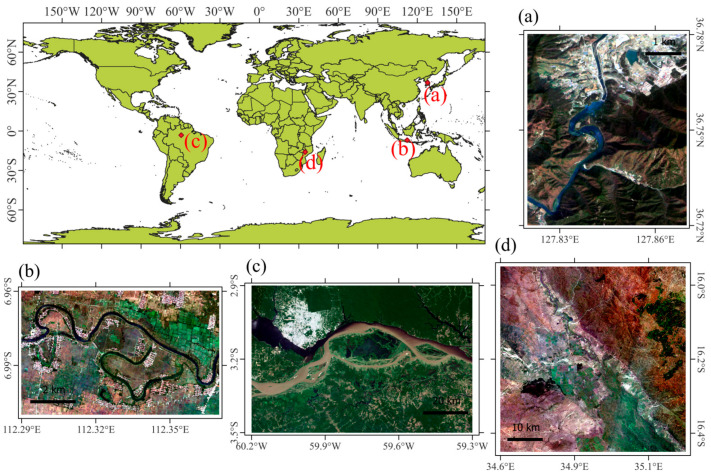
The locations of the study areas and world map images of (**a**) Goesan Dam, Republic of Korea, (**b**) Solo River, Indonesia, (**c**) Amazon River, Brazil, and (**d**) Shire River, Malawi.

**Figure 2 sensors-25-02503-f002:**
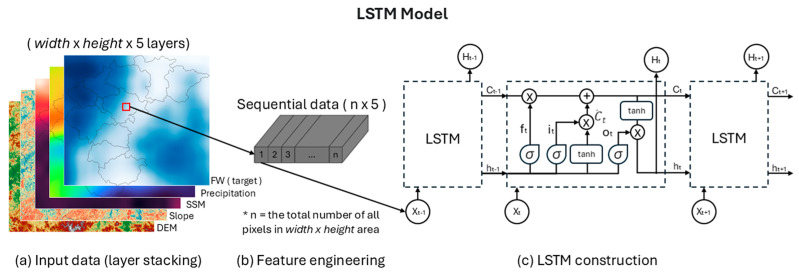
Modeling process of long short-term memory (LSTM) model. (**a**) All input is stacked to form a multi-band image, (**b**) all pixels are extracted and sorted into sequential data and (**c**) sent to the LSTM network.

**Figure 3 sensors-25-02503-f003:**
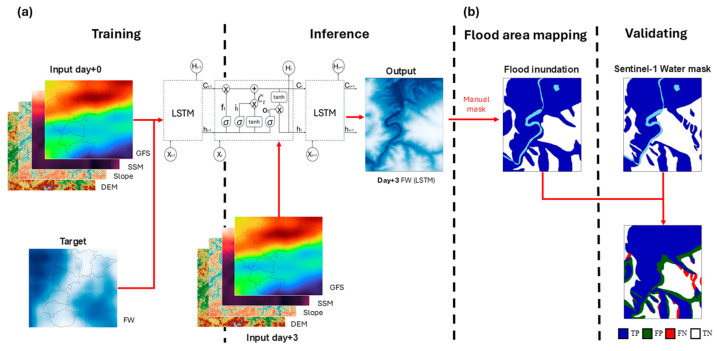
The process of mapping the flood inundation area using LSTM. (**a**) Day+0 and day+3 model training and testing processes and (**b**) model validation process.

**Figure 4 sensors-25-02503-f004:**
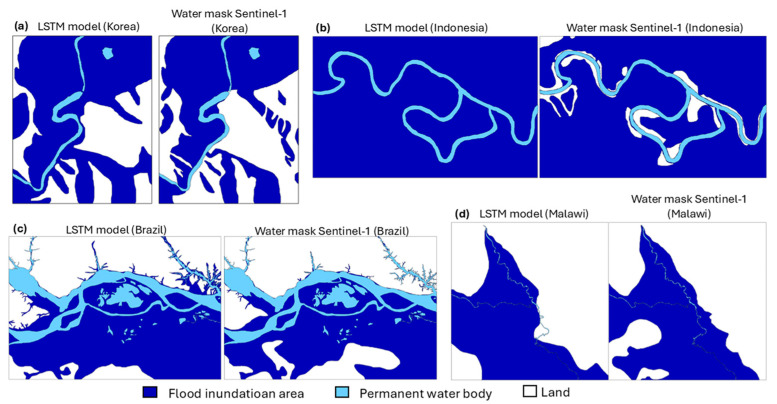
Flood inundation maps from the LSTM model and Sentinel-1 SAR in (**a**) Korea, (**b**) Indonesia, (**c**) Brazil, and (**d**) Malawi.

**Figure 5 sensors-25-02503-f005:**
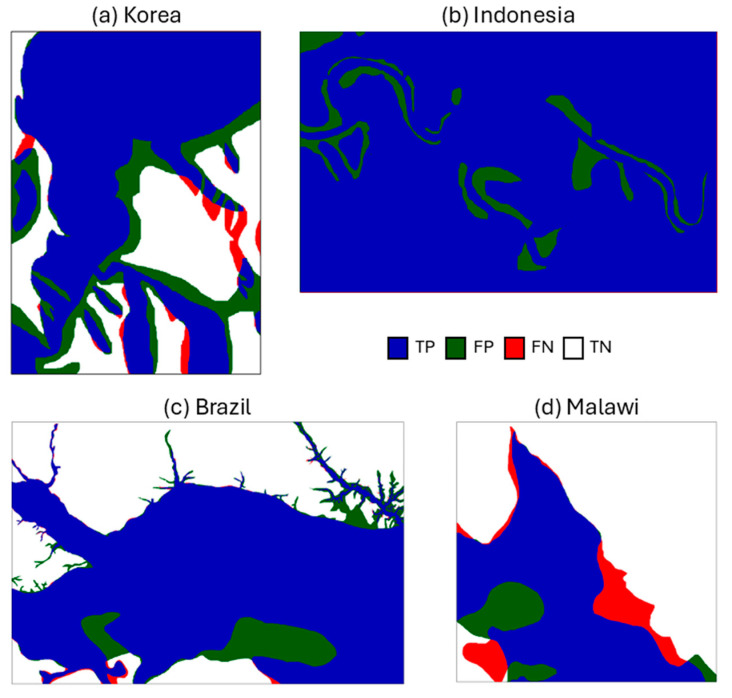
Confusion matrix derived from the water mask of the model and Sentinel-1 imagery.

**Figure 6 sensors-25-02503-f006:**
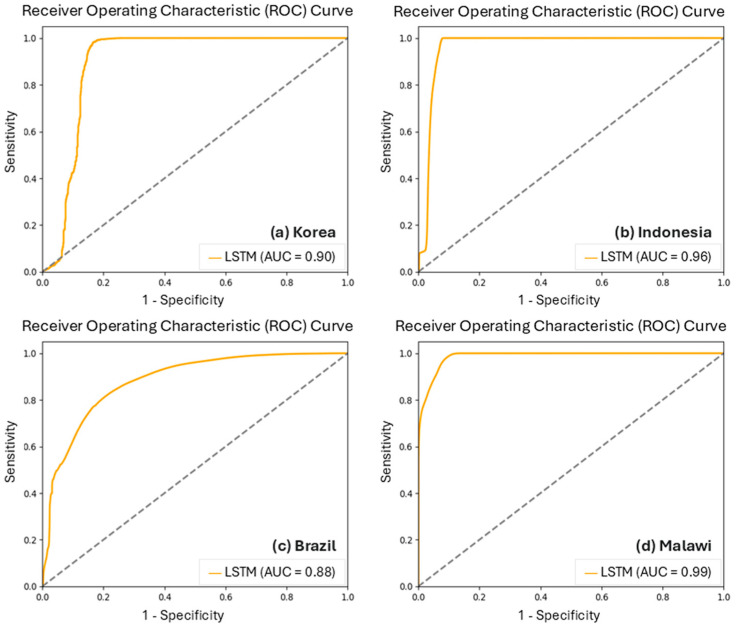
Receiver operating characteristic (ROC) curve of the LSTM model.

**Figure 7 sensors-25-02503-f007:**
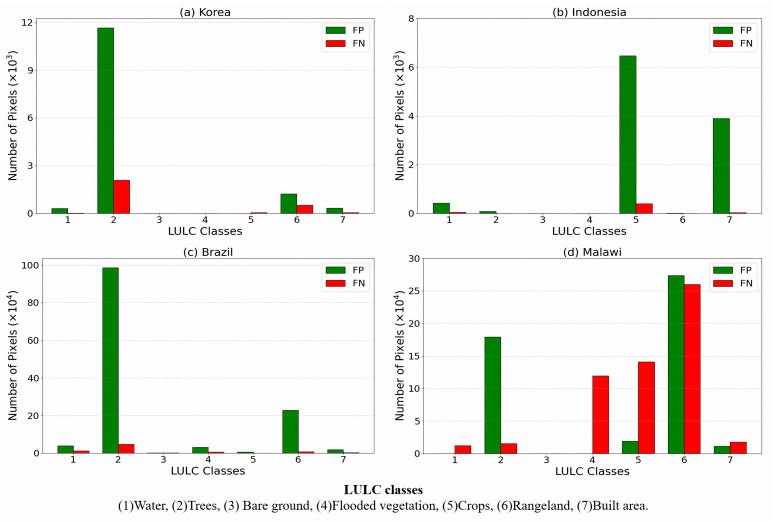
The numbers of pixels in false positive (FP) and false negative (FN) areas based on land use/land cover (LULC) class.

**Figure 8 sensors-25-02503-f008:**
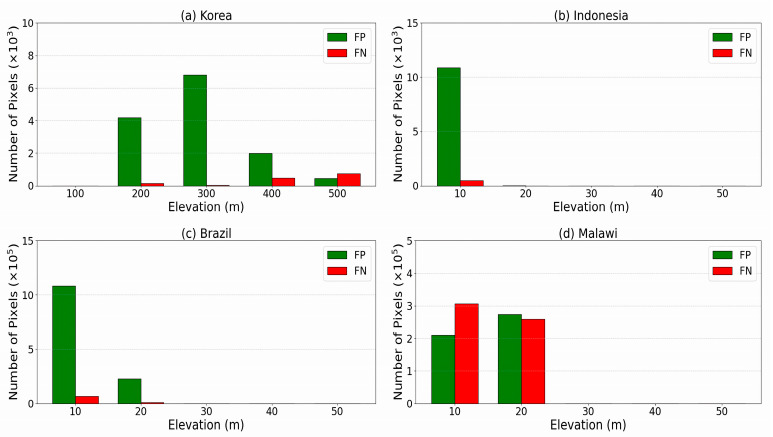
The numbers of pixels in FP and FN areas at different elevations.

**Table 1 sensors-25-02503-t001:** Single-band image sizes and modeling dates for the test sets.

Study Area	Width × Height	Area Extent	Day+0	Day+3
Korea	5007 × 4474	36.01 to 37.26 N, 127.26 to 128.65 E	15 July 2023	18 July 2023
Indonesia	2700 × 4474	−7.96 to −6.91 S, 111.75 to 112.50 E	27 February 2021	2 March 2021
Brazil	4500 × 4500	−3.75 to −2.50 S, −60.50 to −59.25 W	29 April 2021	2 May 2021
Malawi	4500 × 4500	−16.75 to −15.50 S, 34.20 to 35.45 E	23 January 2022	26 January 2022

**Table 2 sensors-25-02503-t002:** Confusion matrix from model and validation.

Study Area	Accuracy	Precision	Recall	IoU	F1 Score
Korea	0.75	0.81	0.91	0.75	0.86
Indonesia	0.91	0.92	0.99	0.91	0.95
Brazil	0.88	0.89	0.99	0.88	0.94
Malawi	0.74	0.86	0.84	0.74	0.85

## Data Availability

Data available in a publicly accessible repository. The original data presented in the study are openly available in NASA EarthData and GEE at https://www.earthdata.nasa.gov/ (accessed on 8 April 2025) and https://developers.google.com/earth-engine/datasets/catalog/NOAA_GFS0P25 (accessed on 8 April 2025).

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
