# Peer review of "Estimation of Flood Inundation Area Using Soil Moisture Active Passive Fractional Water Data with an LSTM Model"

_sensors, 2025, doi:10.3390/s25082503_

Round 1

Reviewer 1 Report

Comments and Suggestions for Authors

Overall, the study demonstrates a promising approach to flood mapping but requires deeper methodological transparency and contextualization of results to maximize its impact. Here are the details:

1.Clarity of Writing and Logical Flow

        The introduction provides a comprehensive overview of flood impacts globally but lacks a focused literature review that clearly identifies research gaps. The motivation for integrating SMAP data with LSTM is not sufficiently contrasted against existing methods (e.g., SAR-only approaches, other ML models). A more structured comparison of prior works would strengthen the rationale for this study.

        The transition between sections (e.g., Methods to Results) is abrupt. For instance, the data preprocessing steps (e.g., bicubic interpolation) are described technically but lack justification for choosing this method over alternatives (e.g., bilinear or nearest-neighbor interpolation). Similarly, the LSTM architecture (e.g., 25 hidden cells) is stated without explaining how hyperparameters were optimized.

2.Methodological Transparency

        While the datasets (SMAP, Sentinel-1, GFS, ASTER-GDEM) are listed, critical details are missing: (1) Temporal resolution of SMAP FW data and how temporal gaps (e.g., 3-day revisit) affect model training. Specific criteria for selecting flood events (Table 1) and whether seasonal variations were considered. (2) Validation metrics (e.g., AUC, F1 score) are defined, but the thresholds for binary classification (water/non-water) are not explicitly stated. This ambiguity affects reproducibility.(3) The LSTM implementation lacks clarity: Input sequence structure (e.g., time steps, feature engineering) and training duration (epochs, loss function) are omitted. The rationale for using day+0 data for training and day+3 for testing needs justification, especially given potential temporal inconsistencies in flood dynamics.

3.Results Presentation

        Figure 2 and Table 2 highlight regional performance disparities (e.g., Malawi vs. Indonesia) but fail to quantitatively link these differences to site-specific factors (e.g., vegetation density, DEM quality). For example, the lower accuracy in Malawi is attributed to terrain complexity but lacks statistical evidence (e.g., correlation between elevation variance and FP/FN rates). The ROC curve (Figure 4) is mentioned but not analyzed in context. How do AUC values compare across regions? Are there thresholds where performance drops significantly?

4.Discussion of Limitations and Implications

        The discussion focuses on DEM and vegetation impacts but overlooks critical limitations:SMAP’s coarse resolution (9 km → resampled to 30 m) introduces uncertainty, yet the paper does not quantify how interpolation errors propagate into flood estimates. The model’s reliance on static DEM data ignores dynamic factors like soil infiltration rates or urbanization changes, which could affect long-term applicability. The practical implications for disaster management are underdeveloped. For instance, how does the 3-day forecast window align with real-world emergency response timelines?

5.Conclusion Specificity

        The conclusion reiterates findings but does not articulate actionable recommendations or future directions. For example, proposing integration with real-time sensor networks or ensemble models (e.g., CNN-LSTM hybrids) would strengthen the manuscript.

Comments on the Quality of English Language

The author can comprehensively polish the language of this article through some large models

Author Response

Comments 1: The introduction provides a comprehensive overview of flood impacts globally but lacks a focused literature review that clearly identifies research gaps. The motivation for integrating SMAP data with LSTM is not sufficiently contrasted against existing methods (e.g., SAR-only approaches, other ML models). A more structured comparison of prior works would strengthen the rationale for this study.

Response: Thank you for your comment. We were motivated to use the SMAP FW data due to capability of this spatial data for flood area determination, so we tried to integrate it with deep learning models to obtain flood inundation area predictions. We have revised the manuscript based on the following information.

Page 2, Line 65 “While optical sensors such as Landsat provide fine resolution (30 m) and SAR systems like Sentinel-1 offer high spatial resolution (10-30 m), both are constrained by relatively long revisit cycles (16 days and 12 days, respectively) [9,10]. Conversely, the Moderate Resolution Imaging Spectroradiometer (MODIS) delivers frequent daily observations (1-2 days) at Moderately fine spatial scales (250 m to 1 km), but its effectiveness is significantly compromised by cloud obstruction during flood events [11,12]. This persistent trade-off among spatial resolution, temporal frequency, and weather penetration capability presents substantial challenges for comprehensive flood monitoring.

Page 3, Line 93 “SMAP data effectively addresses key observational limitations in flood monitoring through its unique capabilities. Unlike optical sensors, which are constrained by cloud cover, or SAR systems, which are limited by their orbital configurations and extended revisit cycles, SMAP provides consistent high-frequency observations through its L-band radiometer [21]. This advanced instrument acquires global FW measurements every 2-3 days, offering reliable, all-weather monitoring capabilities. The system's uninterrupted data stream during flood events represents a significant advancement for hydrological monitoring, overcoming the temporal resolution constraints inherent in both optical and SAR satellite systems.”

Page 3, Line 116 “Conventional machine learning algorithms, including Support Vector Machines (SVM) and Random Forest (RF), have demonstrated competence in processing structured spatial data and offer valuable model interpretability [26]. However, these conventional methods exhibit significant limitations in capturing the critical temporal dependencies inherent in flood dynamics.”

Page 3, Line 130 “A CNN demonstrated exceptional capability in processing spatial data through their hierarchical feature extraction architecture [29]. These networks employ convolutional operations to systematically identify and learn spatial patterns in satellite imagery. However, while CNNs have demonstrated robust performance in static spatial analysis of flood patterns, their fundamental architecture is constrained by an inability to effectively capture and model temporal dependencies and sequential relationships in hydrological data [30].”

References added

Wang, J., Ding, J., Li, G., Liang, J., Yu, D., Aishan, T., ... & Liu, J. (2019). Dynamic detection of water surface area of Ebinur Lake using multi-source satellite data (Landsat and Sentinel-1A) and its responses to changing environment. Catena, 177, 189-201. [9]

Geudtner, D., Torres, R., Snoeij, P., Davidson, M., & Rommen, B. (2014, July). Sentinel-1 sys-tem capabilities and appli-cations. In 2014 IEEE geoscience and remote sensing symposi-um (pp. 1457-1460). IEEE. [10]

Rahman, M. S., Di, L., Yu, E., Lin, L., Zhang, C., & Tang, J. (2019). Rapid flood progress monitoring in cropland with NASA SMAP. Remote Sensing, 11(2), 191. [21]

Ganjirad, M., & Delavar, M. R. (2023). Flood risk mapping using random forest and support vector machine. ISPRS Annals of the Photogrammetry, Remote Sensing and Spatial Infor-mation Sciences, 10, 201-208. [26]

Kattenborn, T., Leitloff, J., Schiefer, F., & Hinz, S. (2021). Review on Convolutional Neural Networks (CNN) in vegetation remote sensing. ISPRS journal of photogrammetry and remote sensing, 173, 24-49. [29]

Mohamadiazar, N., Ebrahimian, A., & Hosseiny, H. (2024). Integrating deep learning, satellite image processing, and spatial-temporal analysis for urban flood prediction. Journal of Hy-drology, 639, 131508. [30]

Comments 2: The transition between sections (e.g., Methods to Results) is abrupt. For instance, the data preprocessing steps (e.g., bicubic interpolation) are described technically but lack justification for choosing this method over alternatives (e.g., bilinear or nearest-neighbor interpolation). Similarly, the LSTM architecture (e.g., 25 hidden cells) is stated without explaining how hyperparameters were optimized.

Response: Thank you for pointing this out. We have added explanations for the data preprocessing steps in the text. We have revised the text as follows.

Page 6, Line 224 “The most used interpolation algorithms today include nearest neighbor, bilinear, and bicubic interpolation, each offering varying levels of accuracy and image quality. Nearest neighbor interpolation is the simplest method, approximating the sinc function in a two-dimensional image. It determines the pixel value of an insertion point by comparing the distances between four surrounding pixels and selecting the nearest one. While computationally efficient, this method often results in blocky artifacts and lower image quality. In contrast, bilinear interpolation improves accuracy by utilizing four surrounding pixels, applying linear interpolation both vertically and horizontally. This approach generates more continuous outputs than nearest neighbor interpolation but still has limitations in accuracy, particularly in high-detail applications. For even greater precision, bicubic interpolation employs sixteen surrounding pixels, significantly enhancing both accuracy and image quality [41]. By considering a larger neighborhood of pixels, bicubic interpolation reduces artifacts and produces more natural transitions, making it preferable for high-resolution image processing and spatial data interpolation.”

References added

Zhou, Z., Wang, Y., Xu, C., & Zhang, Y. (2023, December). Bi-Cubic Interpolation Image Scaling Algorithm Based on FPGA Implementation. In 2023 3rd International Conference on Computer Science, Electronic Information Engineering and Intelligent Control Technology (CEI) (pp. 364-369). IEEE. [41]

Response: Thank you for pointing this out. We have added explanations for the LSTM architecture methods in the text. We have revised the text as follows.

Page 8, Line 298             “In the LSTM modeling process with an aim to minimize loss, the choice of training strategy and hyperparameters critically affects model efficacy. Tanh activation functions were used, a hidden size of 25 was applied, batch sizes of 500 were utilized, the optimizer was set to Adagrad, the loss function was set to MSELoss, and a learning rate of 0.002 was implemented for training from a combination of several sets. An early stop was applied if the development score did not increase for more than 10 training epochs, hyperparameter optimization identified the combination that produced minimal loss, and the highest development score is considered the best.”

Comments 3: While the datasets (SMAP, Sentinel-1, GFS, ASTER-GDEM) are listed, critical details are missing: (1) Temporal resolution of SMAP FW data and how temporal gaps (e.g., 3-day revisit) affect model training. Specific criteria for selecting flood events (Table 1) and whether seasonal variations were considered. (2) Validation metrics (e.g., AUC, F1 score) are defined, but the thresholds for binary classification (water/non-water) are not explicitly stated. This ambiguity affects reproducibility. (3) The LSTM implementation lacks clarity: Input sequence structure (e.g., time steps, feature engineering) and training duration (epochs, loss function) are omitted. The rationale for using day+0 data for training and day+3 for testing needs justification, especially given potential temporal inconsistencies in flood dynamics.

Response (1): Thank you for your constructive comments. In the manuscript we have included temporal resolution from SMAP FW and criteria for selecting flood events. We have added information on and how temporal gaps (e.g., 3-day revisit) affect model training. We have revised the manuscript as follows.

Page 2, Line 89 “The SMAP fractional water (FW) cover generated with 3-day temporal resolution from SMAP TB data has enhanced L-band (1.4 GHz) microwave sensitivity to TB water surface and less affected by atmospheric and vegetation noise than optical/higher-frequency microwave data. Surface FW retrievals de-rived from SMAP L-band TB can be used for flood estimation [18]”

Page 2, Line 45 “Extreme rainfall events and inadequate storage capacity are the main driving factors in flooding events in various regions. The third heaviest rainfall on record, during the 2023 East Asian rainy season, resulted in severe flooding and landslides across Republic of Korea. In Lamongan, Indonesia, heavy rainfall in early March 2021, caused the water discharge in the Solo River watershed to rise, affecting numerous villages on Java Island. In Manaus, Brazil, in the Amazon Basin, a high-water level of 29.19 meters compared to a normal level of 27.00 meters in May 2021, after heavy rainfall, resulted in severe flooding along the river. The impact of Tropical Storm Ana triggered widespread flooding, infra-structure damage, and loss of life throughout Malawi on 26 January 2022.”

Page 8, Line 321 “Flood events often exhibit rapid onset and recession, occurring on timescales shorter than three days. The SMAP dataset may fail to capture critical flood evolution stages, leading to incomplete event representation. Models trained on such data may struggle to accurately predict extreme flood events due to missing key transitional phases. Sequential models, such as LSTM, require consistent time intervals between observations for optimal performance [45]. However, the non-uniform time gaps in SMAP data disrupt the sequential dependencies these models rely on, necessitating data imputation or augmentation techniques to maintain temporal coherence. Without appropriate handling, these discontinuities may degrade model learning and prediction reliability. Furthermore, short-duration flood events, which may last only a few days, are underrepresented in SMAP observations due to the 3-day revisit cycle [20].”

References added

Malik, H., Feng, J., Shao, P., & Abduljabbar, Z. A. (2024). Improving flood forecasting using time-distributed CNN-LSTM model: a time-distributed spatiotemporal method. Earth Science Informatics, 17(4), 3455-3474. [45]

Du, J., Kimball, J. S., Sheffield, J., Pan, M., Fisher, C. K., Beck, H. E., & Wood, E. F. (2021). Satellite flood inundation assessment and forecast using SMAP and landsat. IEEE journal of selected topics in applied earth observations and remote sensing, 14, 6707-6715. [20]

Response (2): Thank you for your constructive comments. We have added information about the stated threshold for binary classification. We have revised the text as follows.

Page 10, Line 358 “Even after a thorough search of the available references, a comprehensive search revealed no established FW-based grading system for surface inundation extent. The classification in this study was determined solely through exploration data assessment. The classification of inundation based on FW values is defined as follows: FW values below 0.15 indicate areas unaffected by flooding, while values of 0.15 or higher denote areas affected by flooding. This classification is derived from preliminary statistical analysis, as there is no widely recognized system for grading inundation levels using FW [46].”

References added

Ma, Z., Zhang, S., Liu, Q., Feng, Y., Guo, Q., Zhao, H., & Feng, Y. (2024). Using CYGNSS and L-band radiometer observations to retrieve surface water fraction: A case study of the catastrophic flood of 2022 in Pakistan. IEEE Transactions on Geoscience and Remote Sensing. [46]

Response (3): Thank you for your constructive comments. We have added information about the implementation of the LSTM model and justification for using day+0 data for training and day+3 for testing. We have revised the text as follows.

Page 8, Line 298 “In the LSTM modeling process with an aim to minimize loss, the choice of training strategy and hyperparameters critically affects model efficacy. Tanh activation functions were used, a hidden size of 25 was applied, batch sizes of 500 were utilized, the optimizer was set to Adagrad, the loss function was set to MSELoss, and a learning rate of 0.002 was implemented for training from a combination of several sets. An early stop was applied if the development score did not increase for more than 10 training epochs, hyperparameter optimization identified the combination that produced minimal loss, and the highest development score is considered the best.”

Page 9, Line 332 “The rationale for using day+0 data for training and day+3 for testing is based on the 3-day temporal resolution of SMAP FW data. SMAP FW provides updates every three days, meaning each observation represents conditions over a multi-day period rather than an instantaneous snapshot. Training on day+0 ensures that the model learns from the most recent and directly observed flood conditions, while testing on day+3 evaluates the model’s predictive capability for future conditions within the constraints of SMAP’s observational cycle. However, flood dynamics can change rapidly and potentially introducing temporal inconsistencies. These inconsistencies must be carefully considered, as flood extents may evolve significantly between SMAP observation windows, impacting model performance.”

Comments 4: Figure 2 and Table 2 highlight regional performance disparities (e.g., Malawi vs. Indonesia) but fail to quantitatively link these differences to site-specific factors (e.g., vegetation density, DEM quality). For example, the lower accuracy in Malawi is attributed to terrain complexity but lacks statistical evidence (e.g., correlation between elevation variance and FP/FN rates). The ROC curve (Figure 4) is mentioned but not analyzed in context. How do AUC values compare across regions? Are there thresholds where performance drops significantly?

Response: Thank you for pointing this out. We have added information about the correlation between elevation variance and FP/FN rates to quantitatively link the accuracy of the model. We have revised the text as follows.

Page 12, Line 427 “The correlation results show that elevation variance has a strong relationship with the FP rate in all sites, with a correlation coefficient of 0.82. This indicates that more complex terrain, with greater elevation variance, contributes to an increase in false positives and leads to lower accuracy. However, the weaker correlation between elevation variance and FN rate (0.21) suggests that other factors, aside from elevation, may play a more significant role in causing false negatives. Therefore, while terrain complexity has a significant impact on FP, the causes of FN are affected by land cover type and other complex environ-mental factors in addition to elevation.

Response: Thank you for pointing this out. We have added information about AUC performance and its analysis.

Page 12, Line 450 “The ROC curve analysis of the LSTM model across Korea (AUC = 0.90), Indonesia (AUC = 0.96), Brazil (AUC = 0.88), and Malawi (AUC = 0.99) demonstrates strong classification performance, with particularly high accuracy in Malawi and Indonesia. While Korea and Brazil show slightly lower AUC values, they still indicate robust predictive capabilities, this variation is likely attributable to regional topographic heterogeneity, hydrological variability, and dataset quality. Brazil’s diverse flood-prone landscapes and Korea’s varied elevation may intro-duce additional challenges in feature learning. Despite these variations, the model maintains stable performance across all regions, achieving a favorable balance between true positive and false positive rates. The absence of an abrupt threshold where performance drops significantly suggests reliable generalization across different hydrological conditions. However, for applications requiring precise trade-offs between sensitivity and specificity, further threshold tuning may be necessary. Overall, the consistently high AUC values underscore the model's reliability for cross-regional flood prediction.”

Comments 5: The discussion focuses on DEM and vegetation impacts but overlooks critical limitations: SMAP’s coarse resolution (9 km → resampled to 30 m) introduces uncertainty, yet the paper does not quantify how interpolation errors propagate into flood estimates. The model’s reliance on static DEM data ignores dynamic factors like soil infiltration rates or urbanization changes, which could affect long-term applicability. The practical implications for disaster management are underdeveloped. For instance, how does the 3-day forecast window align with real-world emergency response timelines?

Response: Thank you for your constructive comments. The process of downscaling SMAP FW data from 9 km resolution to 30 m presents significant challenges in error quantification and propagation analysis. These difficulties come from two fundamental issues: (1) the inherent non-linear scaling effects introduced by interpolation methods, and (2) the absence of reliable high-resolution reference data with matching spatiotemporal characteristics for proper validation. The bicubic interpolation process, while mathematically robust, introduces complex error patterns that are particularly difficult to characterize.

The study "Satellite Flood Inundation Assessment and Forecast Using SMAP and Landsat" (Du et al., 2021) demonstrates the synergistic use of SMAP (9 km) and Landsat (30 m) data for flood monitoring. The study employs an interpolation method to distribute SMAP FW values to 30 m pixels. While this approach improves spatial consistency, it does not quantify how errors from the original SMAP data propagate through the interpolation process.

We believe recent advances in AI offer promising approaches to address these challenges. These AI approaches do not eliminate interpolation uncertainty entirely, but they provide a probabilistic framework for understanding and mitigating errors. And we have revised the manuscript as follows.

Page 13, Line 477 “The referenced study [20] successfully integrated dataset for flood monitoring but did not quantify how errors from the original SMAP data propagate through the interpolation process into 30m, potentially affecting the accuracy of downscaled flood maps. While empirical interpolation improves spatial resolution, it may introduce unaccounted-for un-certainties. The challenges primarily originate from fundamental limitations in current methodologies, particularly the non-linear distortions introduced by conventional interpolation techniques and the critical absence of high-resolution validation datasets.”

References added

Du, J., Kimball, J. S., Sheffield, J., Pan, M., Fisher, C. K., Beck, H. E., & Wood, E. F. (2021). Satellite flood inundation assessment and forecast using SMAP and landsat. IEEE journal of selected topics in applied earth observations and remote sensing, 14, 6707-6715. [20]

Response: Thank you for your comments. We have added information about the 3-day forecast window align with real-world emergency response timelines.

Page 14, Line 506 “The 3-day forecast window is crucial in emergency response because it provides enough time for preparedness while maintaining forecast accuracy. If the window is too short, such as less than 24 hours, authorities may struggle to issue timely warnings, and large-scale evacuations could become chaotic or even impossible. On the other hand, forecasts extending beyond five days tend to be less reliable, leading to wasted resources and inefficient planning. By striking the right balance, a 3-day forecast allows emergency teams to mobilize resources, issue alerts, and implement safety measures effectively, ensuring communities are well-prepared while minimizing unnecessary disruptions.”

Comments 6: The conclusion reiterates findings but does not articulate actionable recommendations or future directions. For example, proposing integration with real-time sensor networks or ensemble models (e.g., CNN-LSTM hybrids) would strengthen the manuscript.

Response: Thank you for your comments. We have added actionable recommendations or future directions. We have revised the manuscript as follows.

Page 16, Line 594 “Since the current LSTM model uses only a single time step, its ability to capture long-term trends is limited. Enhancing the model with multi-step sequences, changing FW data using finer spatial resolution data from other L-band satellites (CYGNSS) with a spatial resolution of 3 km to reduce the uncertainties, incorporating external hydrological data, or integrating convolutional layers (CNN-LSTM hybrid) could further improve predictive performance.”

Reviewer 2 Report

Comments and Suggestions for Authors
  1. In the Methodology section of the manuscript, the flowchart of the research methods is a crucial part, as it helps readers quickly understand the data processing and modeling process. Therefore, it is recommended to retain Figures S1 and S2 from the supplementary materials in the main manuscript. Additionally, there are some issues related to these figures and their description in the text that need to be clarified to improve the reproducibility of the article: 1) In the data processing step, how is the image data input into the LSTM? When stacking the data, was the 2D image data flattened, as LSTM is not particularly adept at handling spatial data? 2) What are the inputs and outputs of the LSTM, and what are their dimensions (i.e., how is n determined)? 3) The manuscript mentions that the day+0 data is used for training, but why does Figure S2 show FW as part of the input? Ihope the authors can address these issues in Section 4.3 of the Methodology and further refine the expressions in Figures S1 and S2.

  2. As LSTMadept at handling sequential data, when applying LSTM to spatial geoinformation data modeling and analysis, how can it account for the interaction between input data points (such as the relationship between a DEM raster and its surrounding rasters)? If this is not the case, the application of LSTM might pose risks to the physical consistency and interpretability (reliability) of the results.

  3. In line 153, in Table 2, please clarify the actual meaning of w×h.

  4. It is recommended to add references to the data sources for the datasets mentioned in lines 163, 167, 179, 185, and 195.

  5. In line 205, the content "(Fang et al., 2021)" seems redundant and should be removed.

  6. In line 171, the variables' superscripts and subscripts should be formatted in the equation format. In Equations 2, 3, 4, 5, 6, 7, and 8, please ensure that terms like 0, 0+i, t, t-1, etc., are adjusted to appear as subscripts.

  7. Figure 4 presents the ROC results of the LSTM model. It is recommended to define ROC prior to its use for evaluation.

Comments on the Quality of English Language

The English could be improved to more clearly express the research.

Author Response

Comments 1: In the Methodology section of the manuscript, the flowchart of the research methods is a crucial part, as it helps readers quickly understand the data processing and modeling process. Therefore, it is recommended to retain Figures S1 and S2 from the supplementary materials in the main manuscript. Additionally, there are some issues related to these figures and their description in the text that need to be clarified to improve the reproducibility of the article: 1) In the data processing step, how is the image data input into the LSTM? When stacking the data, was the 2D image data flattened, as LSTM is not particularly adept at handling spatial data? 2) What are the inputs and outputs of the LSTM, and what are their dimensions (i.e., how is n determined)? 3) The manuscript mentions that the day+0 data is used for training, but why does Figure S2 show FW as part of the input? I hope the authors can address these issues in Section 4.3 of the Methodology and further refine the expressions in Figures S1 and S2.

Response: Thank you for pointing this out. We have added Figures S1 and S2 (Figure 2 and Figure 3) to the manuscript. We have revised the manuscript as follows.

Page 7, Line 253 “Figure 2. Modeling process of LSTM. (a) All input is stacked to form a multi-band image, (b) all pixels are extracted and sorted into sequential data and (c) sent to the LSTM network.”

Page 9, Line 343 “Figure 3. The process of mapping the flood inundation area using LSTM. (a) Day+0 and day+3 model training and testing processes and (b) model validation process.”

Response (1): Thank you for your constructive comments. In our DL workflow, the 2D image data is not fully flattened into a 1D vector. Instead, the data is processed at the individual pixel level, which helps to maintain the spatial structure within each pixel. For every pixel, feature values are extracted from multiple raster layers, capturing a variety of environmental characteristics. This method effectively transforms the image data into a pixel-wise structured dataset, where each pixel is treated as a separate sample. By organizing the data in this way, the model is able to process each pixel individually while preserving the integrity of the original spatial features across different raster layers. This structure enables the model to handle complex, multi-dimensional data while maintaining a consistent framework for analysis. We have added information about data processing step. We have revised the manuscript as follows.

Page 8, Line 287 “The 2D image data is processed at the pixel level, preserving spatial structure by extracting multi-layer feature values for each pixel and effectively transforming the image into a pixel-wise structured dataset. This approach enables independent pixel analysis while maintaining spatial integrity across raster layers, ensuring robust handling of complex multi-dimensional environmental data.”

Response (2,3): Thank you for pointing this out. We have added information about data processing step. We have revised the manuscript as follows.

Page 8, Line 307 “LSTM model first processes FW as input and output target variables and treated geospatial raster data from DEM, SSM, and meteorological variables, representing various environmental and hydrological as predictors. These raster datasets are normalized using min-max scaling to ensure on a comparable scale from all features, preventing dominance by large-value variables. Reformats pixel-based data into sequential input samples, ensuring that the LSTM can capture temporal dependencies and learns temporal patterns from the geospatial predictors to predict floodwater presence at each pixel. The number of pixels (n) from Figure 2 that are processed is equal to the number of values width x height in the Table 1.”

Comments 2: As LSTM adept at handling sequential data, when applying LSTM to spatial geoinformation data modeling and analysis, how can it account for the interaction between input data points (such as the relationship between a DEM raster and its surrounding rasters)? If this is not the case, the application of LSTM might pose risks to the physical consistency and interpretability (reliability) of the results.

Response: Thank you for your constructive comments. To enable LSTM to process spatial dependencies, raster-based geospatial data is reformatted into sequential input samples. This is achieved by treating each pixel and structuring its values alongside those of its neighboring pixels. The model then learns spatial dependencies in a pseudo-temporal manner, capturing patterns similar to how it would. Instead of processing pixels independently, the model considers a moving window. This allows LSTM to incorporate information from surrounding pixels, effectively capturing spatial interactions within the dataset. Batch normalization is applied to standardize inputs across mini-batches, ensuring consistent scaling and improving model stability. This helps LSTM handle variations in geospatial predictors (e.g., differences in environmental and hydrological conditions) without being biased toward specific regions. This enables the model to generalize across different flood scenarios and predict flood occurrence at each pixel with higher accuracy.

Page 8, Line 291 “Raster-based geospatial data is restructured into sequential samples, where each pixel's values are processed alongside its neighbors within a moving window and identifying temporal patterns. Batch normalization is applied to standardize inputs, enhancing model stability and reducing regional biases across varying conditions. This approach allows LSTM to effectively learn spatial interactions, improving generalization and predict accuracy for flood occurrence at the pixel level.”

Comments 3: In line 153, in Table 2, please clarify the actual meaning of w×h.

Response: Thank you for your comment. We clarified the actual meaning of width × height.

Page 5, Line 176 “Study area      Width x Height   Area extent         Day+0  Day+3.”

Comments 4: It is recommended to add references to the data sources for the datasets mentioned in lines 163, 167, 179, 185, and 195.

Response: Thank you for your comment. We just added some references to the section because we tried to collect data from different sources.

Page 5, Line 183 (Example) “This framework simulates soil water transfer between surface layers and deeper root zones, generating global SSM and root zone moisture fields at enhanced 9 km spatial res-olution from https://search.earthdata.nasa.gov/search [34].”

References added

Duygu, M. B., & Akyürek, Z. (2019). Using cosmic-ray neutron probes in validating satellite soil moisture products and land surface models. Water, 11(7), 1362. [34]

Miao, X., Wang, Y., Yang, Y., & Li, H. (2018, July). Estimating Hi-Resolution Soil Moisture Da-ta Using the HP Model Coupled with Landsat-8 and Smap Datasets. In IGARSS 2018-2018 IEEE International Geoscience and Remote Sensing Symposium (pp. 9106-9109). IEEE. [35]

Yan, D., Wang, K., Qin, T., Weng, B., Wang, H., Bi, W., ... & Abiyu, A. (2019). A data set of global river networks and corresponding water resources zones divisions. Scientific data, 6(1). [38]

Comments 5: In line 205, the content "(Fang et al., 2021)" seems redundant and should be removed.

Response: Thank you for your comment. We have removed redundant content from the manuscript.

Page 6, Line 239 “To be processed in the LSTM neural network and obtain flood inundation area from FW data, the datasets need to be adjusted to a set resolution and pixel area [42]”

Comments 6: In line 171, the variables' superscripts and subscripts should be formatted in the equation format. In Equations 2, 3, 4, 5, 6, 7, and 8, please ensure that terms like 0, 0+i, t, t-1, etc., are adjusted to appear as subscripts.

Response: Thank you for your comment. We have adjusted the terms to appear as subscripts.

Page 7, Line 261 (Example) “                                   (3)

Comments 7: Figure 4 presents the ROC results of the LSTM model. It is recommended to define ROC prior to its use for evaluation.

Response: Thank you for your comment. We have revised the text as follows.

Page 13, Line 468 “Figure 6. Receiver Operating Characteristics (ROC) curve of the LSTM model.”

Round 2

Reviewer 1 Report

Comments and Suggestions for Authors

The author has clearly taken my suggestions seriously and provided revisions. I agree in principle with the publication of this paper, but there are still several areas requiring improvement:
(1) In the introduction, the author mentions several typical flood events. It is recommended to supplement corresponding citations here and elsewhere in the text. This will both ensure the arguments are well-supported and allow readers to explore these cases in depth.
(2) All abbreviations should be spelled out in full when first appearing in the text. Particularly for abbreviations in figures, explicit explanations must be provided in the figure captions to avoid requiring readers to interpret them from the graphics.
(3) The resolution of all figures is currently too low. Please strictly follow journal formatting requirements for modification. Special attention should be paid to the scale bars in composite images.
(4) While the paper's core contribution lies in using microwave radiometer data for flood extraction, employing neural networks to integrate multi-source data for simulation, and validating with Sentinel data, some concerns remain:
â‘ The significant resolution discrepancies between multi-source data (even after downscaling) require clearer justification for result validityï¼›
â‘¡SMAP data is typically used for large-scale studies, whereas this paper's simulation scale appears relatively small
Essential revision: Add a technical flowchart detailing the application process of each dataset and their interrelationships.
(5)The discussion lacks depth and formal conclusions. Although section 5.4 properly incorporates existing research showing SMAP's better performance in dry, low-biomass areas with seasonal water variations and flat terrain, the following need enhancement:
â‘ Maintain consistency across abstract, technical flowchart, discussion, and conclusions.
â‘¡Strengthen the critical analysis of result limitations.
â‘¢Highlight technical innovations and practical implications.

Reviewer 2 Report

Comments and Suggestions for Authors

This is an article with strong innovation and practical significance. The author has made revisions according to our requirements, please correct minor methodological errors and text editing. 
The article has explored the feasibility and expanded the applicability of LSTM in the field of flood inundation range. In addition, the article puts forward insights into the impact of the underlying surface on the flood inundation range. Overall, this article has certain publication value.
